# Single Infrared Image Stripe Removal via Residual Attention Network

**DOI:** 10.3390/s22228734

**Published:** 2022-11-11

**Authors:** Dan Ding, Ye Li, Peng Zhao, Kaitai Li, Sheng Jiang, Yanxiu Liu

**Affiliations:** 1College of Physics, Changchun University of Science and Technology, Changchun 130022, China; 2College of Electronic Information Engineering, Changchun University, Changchun 130022, China

**Keywords:** infrared image, non-uniformity correction, multi-scale feature, similarity metric, attention mechanism

## Abstract

The non-uniformity of the readout circuit response in the infrared focal plane array unit detector can result in fixed pattern noise with stripe, which seriously affects the quality of the infrared images. Considering the problems of existing non-uniformity correction, such as the loss of image detail and edge blurring, a multi-scale residual network with attention mechanism is proposed for single infrared image stripe noise removal. A multi-scale feature representation module is designed to decompose the original image into varying scales to obtain more image information. The product of the direction structure similarity parameter and the Gaussian weighted Mahalanobis distance is used as the similarity metric; a channel spatial attention mechanism based on similarity (CSAS) ensures the extraction of a more discriminative channel and spatial feature. The method is employed to eliminate the stripe noise in the vertical and horizontal directions, respectively, while preserving the edge texture information of the image. The experimental results show that the proposed method outperforms four state-of-the-art methods by a large margin in terms of the qualitative and quantitative assessments. One hundred infrared images with different simulated noise intensities are applied to verify the performance of our method, and the result shows that the average peak signal-to-noise ratio and average structural similarity of the corrected image exceed 40.08 dB and 0.98, respectively.

## 1. Introduction

Infrared imaging technology has been widely applied in military and civilian fields, such as night vision, surveillance systems, fire detection and robotics [1,2]. However, due to the limitations of the detector material and the manufacturing process, the nonuniformity of the infrared focal plane array unit response typically manifests vertical stripe fixed-pattern noise (FPN) [3,4]. Such FPN is especially obvious in uncooled long-wave infrared imaging systems, and seriously reduces the image quality [5,6]. Consequently, in order to improve the infrared image quality, it is necessary to develop an effective non-uniformity correction (NUC) method to remove the stripe noise.

Over recent decades, lots of NUC methods have been proposed, which can be mainly divided into two categories: calibration-based methods and scene-based methods [7,8]. The calibration-based methods require a uniform radiation source (such as a blackbody) to obtain correction parameters to compensate for non-uniformity, which gives the detector a consistent response at the same temperature. Although the calibration methods are simple, the correction parameters cannot be updated in real time, requiring periodic correction [9,10]. In contrast, the scene-based methods can adaptively alleviate FPN fluctuation through scene information without a uniform radiation source, resulting in correction parameters that can be updated in real time [11]. In general, scene-based methods include multi-frame and single-frame methods [12]. The multi-frame methods that rely on inter-frame scene motion are prone to ghosting artifacts, so they could converge in a specific frame. The single-frame methods, including traditional methods and deep learning methods, have the advantage of fast convergence and almost no ghosting artifacts. The methods based on deep learning have good adaptability and anti-noise ability, while the traditional methods will lead to edge blur [13]. Deep-learning approaches are currently the main research directions to address the problem of infrared image quality, so the NUC methods based on deep learning have been actively proposed [14]. Kuang et al. presented a convolutional neural network (SNRCNN) for single infrared image-stripe noise removal that treats the de-striping task as image denoising and super resolution [15]. He et al. introduced a residual deep network-based NUC method (DLS-NUC) that seeks better de-striping results by learning to compute residual information [16]. Xiao et al. proposed an ICSRN model for a deep convolutional network, utilizing a local–global combination structure to optimize the edge-preserving performance [17]. Lee et al. designed a dual-branch structure stripe removal network to extract the structural features of FPN. The parametric FPN model is used to generate training data [18]. Xu et al. eliminated the stripe artifacts with a deep dense connection convolutional neural network, which extracts the image features at different scales [19].

However, the above-mentioned NUC methods still have a number of limitations, such as ghosting artifacts and blurred edges. During NUC, the infrared image with rich details easily loses details, and an image with dense stripe information is liable to leave noise. In the process of image feature extraction, only the local shallow feature is focused, and the global high-level feature is ignored.

To overcome these limitations, this paper proposes an innovative NUC method based on the attention mechanism and residual network. The raw infrared image is input into the residual network to extract the stripe properties. First of all, a new multi-scale feature extraction (MFE) network is designed to better display the texture information of different scales in the image. After that, the proposed similarity metric method is introduced into the channel spatial attention mechanism. According to the similarity between the feature maps, the stripe information is highlighted in different degrees in channel and space, and the global properties are extracted. The combination of the MFE and the attention mechanism can capture deeper feature relationships and effectively extract stripe features. Ultimately, the estimated stripe information is subtracted from the raw image, and the scene details and FPN are separated to obtain the NUC result.

The major ideas and contributions of the paper are summarized as follows:In view of the phenomena of information loss and noise residue, this paper composes images with diverse noise intensities into a training set, directly learns the stripe property from the image, and precisely and adaptively estimates the noise strength and distribution, yielding superior stripe removal performance.To avoid ghosting artifacts and blurring edges, this paper designs an MFE network to extract stripe features in images at different scales. This structure expands the receptive field while reducing the network parameters, and utilizes the complementarity of different features to improve the accuracy of the NUC.For the problem of ignoring global information in feature extraction, this paper proposes a channel spatial attention mechanism based on similarity (CSAS). Through the similarity between feature maps in channel and space, various degrees of weighting are carried out to extract global features, so as to enhance the internal relationship and highlight meaningful information.

The remainder of the paper is organized as follows: in Section 2, the theoretical principle of the proposed method is introduced. In Section 3, the effectiveness of the network structure is analyzed, and infrared images with respectively simulated and real noise are chosen to verify experimentally their performance by using different correction methods. Finally, conclusions are given in Section 4.

## 2. The Proposed NUC Method

### 2.1. Network Architecture

In this paper, the residual learning strategy is introduced by adding a skip connection between the input and output to seek the estimated non-uniform noise from a noisy input image [20]. The architecture of the proposed method is exhibited in Figure 1. The network mainly consists of three parts: feature extraction, feature enhancement and feature reconstruction.

#### 2.1.1. Feature Extraction

This part is responsible for initial feature extraction and feature map acquisition with only one convolutional layer.

The traditional convolution layer is applied to transform the input image into a feature map with multiple channels in an order that extracts the primary features and prepares for the follow-up work. Given the input image $I_{input}$, we can get the shallow feature $F_{0}$ through the convolution layer $f_{\mathrm{conv}}^{3\times 3.64}$ with a kernel size of 3 × 3.64.
(1)$$\begin{array}{c} F_{0}=f_{conv}^{3\times 3.64}\left(I_{input}\right) \end{array}$$

#### 2.1.2. Feature Enhancement

Then, the extracted feature $F_{0}$ is sent to the part of feature enhancement for deep feature learning. The part is made up of 4 stripe feature extraction modules (SFE) $f_{SFE}$, which can be formulated as
(2)$$\begin{array}{c} F_{1}=f_{SFE}f_{SFE}f_{SFE}f_{SFE}\left(F_{0}\right) \end{array}$$
where $F_{1}$ denotes the output feature after feature enhancement. Furthermore, SFE includes MFE and CSCA, extracting stripe features by image similarity.

#### 2.1.3. Feature Reconstruction

The multi-channel information is fused by convolution layer to reconstruct the stripe noise.
(3)Inoise^=fconv3×3F1
where Inoise^ is the reconstructed stripe noise. $f_{\mathrm{conv}}^{3\times 3}$ indicates a convolution operation with the filter size of 3 × 3.

Finally, the output image $I_{output}$ is calculated by subtracting the reconstructed stripe noise Inoise^ from the input image $I_{input}$ as
(4)Ioutput=Iinput−Inoise^

In addition, to keep the input and output dimensions consistent, we set the padding and stride attribute of convolution operation to be $1/2\left(k-1\right)$ and 1, respectively; k represents the size of the convolution kernel. The filter size of the network is restricted to 3 × 3, as it has been proved that decomposing a larger scale filter into multiple smaller scale filters will make the network more nonlinear. The number of the filter channels of the first and last convolution layers is the same as the number of input infrared image channels.

Except for the first and last convolutional layers, all convolutional layers are followed by a batch normalization (BN) [21] and a rectified linear unit (ReLU) [22]. Because the stripe noise simulated in the training stage has negative information, ReLU is not used in the first layer. If ReLU is used, some residual information will be lost, which will grow the difficulty of predicting the residual image.

### 2.2. Multi-Scale Feature Extraction

As illustrated in Figure 2, the designed MFE is inspired by Inception-ResNet [23] architecture that decomposes the input image into multi-scale representations using filters of different sizes. Stripe features are extracted from these multi-scale representations. We use cascades of the 1 × 1 and 3 × 3 sized filters instead of a single big filter. The purpose of this operation is to reduce the number of parameters and pick effectively shallow features. The use of wider kernels can increase the receptive field of the network. Additionally, ResNet accelerates Inception training, which avoids the diminishing feature reuse that comes with the increase in the number of parameters in the network.
(5)$$\begin{array}{c} F_{1.1}=f_{MFE}\left(F_{0}\right) \end{array}$$
$f_{MFE}$ denotes MFE operation.

### 2.3. Similarity Metric

#### 2.3.1. Gaussian Weighted Mahalanobis Distance

Mahalanobis distance is usually used to calculate the similarity between image blocks [24]. By normalizing the data of each image block, the interference of correlation between pixels is eliminated. The Mahalanobis distance $d\left(i,j\right)$ between two points i and j is presented by
(6)$$\begin{array}{c} d\left(i,j\right)=\sqrt{{\left(i-j\right)}^{T}S^{-1}\left(i-j\right)} \end{array}$$
where S is the overall covariance matrix. The neighborhoods of point i in image X and point j in image Y are expressed as $N_{Xi}$ and $N_{Yj}$, respectively.

For measuring the similarity of two pixels, the Gaussian weighted Mahalanobis distance between these two points can be expressed by
(7)$$\begin{array}{c} d\left(i,j\right)=\|G_{\alpha }•{\left(N_{Xi}-N_{Yj}\right)}^{T}S^{-1}\left(N_{Xi}-N_{Yj}\right){\|}_{2} \end{array}$$
where $G_{\alpha }$ denotes the Gaussian kernel function with standard deviation α. The symbol • denotes dot product; that is, the corresponding elements in the image block are multiplied. $G_{\alpha }$ is used to improve the accuracy of the similarity metric of image blocks, and to reduce the interference of noise in the calculation of the Gaussian weighted Mahalanobis distance.

#### 2.3.2. Direction Structure Similarity Algorithm

As a full-reference image similarity metric, the structural similarity algorithm (*SSIM*) estimates from three different factors: brightness, contrast and structure [25,26]. The formula of SSIM between two image blocks X and Y of size m×m is depicted as follows
(8)$$\begin{array}{c} SSIM\left(X,Y\right)=\frac{\left(2\mu_{X}\mu_{Y}+c_{1}\right)\left(2\sigma_{XY}+c_{2}\right)}{\left(\mu_{X}^{2}+\mu_{Y}^{2}+c_{1}\right)\left(\sigma_{X}^{2}+\sigma_{Y}^{2}+c_{2}\right)} \end{array}$$
where $\mu_{X}$ is the mean value of X, $\mu_{Y}$ is the mean value of Y, $\sigma_{X}$ is the variance of X, $\sigma_{Y}$ is the variance of Y and $\sigma_{XY}$ is the covariance of X and Y. $c_{1}={\left(0.01e\right)}^{2}$ and $c_{2}={\left(0.03e\right)}^{2}$ are coefficients used to maintain stability. e is the dynamic range of pixel values.

Measuring the similarity between image blocks is different from pixels. The image block contains direction information, and the parameters of *SSIM* are based on the gray value of the pixel, which does not reflect the direction structure of the image blocks themselves. Thus, combining the direction structure information and the geometry structure information of the image block can more accurately measure the similarity between the image blocks.

When extracting the direction information of image blocks, the neighborhood $N_{i}$, of the pixel i in the image is divided into two parts, $N_{i\theta 1}$ and $N_{i\theta 2}$, by a straight line with an angle of θ passing through point i. The direction information of point i is the corresponding direction when parameter h takes the maximum value.
(9)$$\begin{array}{c} h=max\left|v_{N_{i\theta 1}}-v_{N_{i\theta 2}}\right| \end{array}$$

Among them, 0°≤θ≤180°, $v_{N_{i\theta 1}}$ and $v_{N_{i\theta 2}}$ are the gray value sum of pixels in $N_{i\theta 1}$ and $N_{i\theta 2}$, separately. In the counterclockwise direction, θ takes as 0°, 45°, 90°, 135°, 180°, 225°, 270°, 315°, respectively. By Formula (9), the difference in grayscale distribution within the pixel neighborhood is calculated. The larger the value h, the greater the difference in pixel grayscale distribution on both sides of the direction line. Therefore, the Formula (9) can effectively reflect the direction information of the image block where point i is located.

The total number of pixels in the image block is a, and the number of pixels with the same direction information is d. The direction information of the pixels at the corresponding positions of the two image blocks X and Y is extracted to be compared. Then, the Formula (8) can be written as
(10)$$\begin{array}{c} SSIM{\left(X,Y\right)}^{\prime }=\frac{\left(2\mu_{X}\mu_{Y}+c_{1}\right)\left(2\sigma_{XY}+c_{2}\right)d}{\left(\mu_{X}^{2}+\mu_{Y}^{2}+c_{1}\right)\left(\sigma_{X}^{2}+\sigma_{Y}^{2}+c_{2}\right)a} \end{array}$$

#### 2.3.3. Improved Similarity Metric

The product $SM{\left(X,Y\right)}^{\prime }$ of the $SSIM{\left(X,Y\right)}^{\prime }$ and the Gaussian weighted Mahalanobis distance is applied to measure the neighborhood block similarity.
(11)$$\begin{array}{c} SM{\left(X,Y\right)}^{\prime }=SSIM{\left(X,Y\right)}^{\prime }\times (1-\|G_{\alpha }•{\left(N_{Xi}-N_{Yj}\right)}^{T}S^{-1}\left(N_{Xi}-N_{Yj}\right){\|}_{2}) \end{array}$$
Here, $SM{\left(X,Y\right)}^{\prime }$ takes a value between −1 and 1; the image has a higher degree of similarity when the absolute value of $SM{\left(X,Y\right)}^{\prime }$ is close to 1.

### 2.4. Attention Mechanism

Attention weights each element of the feature maps to suppress unnecessary ones and only focus on important ones in order to boost the representation power of the network architecture. Similar features would be related to each other. It is necessary to selectively emphasize interdependent feature blocks according to the similarity. Thus, a CSAS that refines and extracts the stripe features more precisely is proposed. The structure of CSAS is illustrated in Figure 3.

#### 2.4.1. Image Block Division

In order to achieve a better denoising effect, 7 × 7 pixels image blocks are selected. In terms of the images whose length or width cannot be divided exactly, the blank part should be expanded for exact division. As can be seen in Figure 4, an infrared image with 640 × 480 pixels is divided into image blocks (70 × 70 pixels in each block) (Figure 4a,b), where the right and bottom edges of the image in Figure 4a are expanded mirror-symmetrically to fill the blank pixels.

#### 2.4.2. Channel Attention Mechanism

In the deep feature map, the semantic features of different channel maps are associated with each other. Each channel is reconstructed by calculating the correlation between channels. The more similar the channels are, the greater the weight assigned and the more important the channels are.

The original feature map $F_{1.1}$ is divided into n feature blocks $B_{p}$ with size of 7 × 7 × 64.
(12)$$\begin{array}{c} B_{p}=block\left(F_{1.1}\right),\left(p=1,2,\ldots ,n\right) \end{array}$$
block represents grouping operation, $B_{p}$ indicates the $p^{\mathrm{th}}$ group feature block.

The similarity is calculated between 64 channels in $B_{p}$ to obtain a 64 × 64 channel similarity matrix. The channel similarity matrix is normalized by sigmoid activation function to get the channel weight matrix $W_{p}^{c}$. This simulates the dependencies between channels and helps to boost feature extraction capability.
(13)$$\begin{array}{c} W_{p}^{c}=softmax\left(SM^{\prime }\left(B_{p}\right)\right) \end{array}$$
$B_{p}$ can be regarded as a matrix of 1 × 64; $B_{p}$ and $W_{p}^{c}$ are multiplied to obtain n groups of new feature blocks $B_{p}^{\prime }$. The symbol × denotes cross-product.
(14)$$\begin{array}{c} B_{p}^{\prime }=B_{p}\times W_{p}^{c} \end{array}$$

#### 2.4.3. Spatial Attention Mechanism

Spatial attention mechanism focuses on the information region of the spatial dimension and emphasizes contextual information. We obtain the weight by calculating the similarity between image blocks in each channel, which enhances or weakens the feature at each position.

$B_{p}^{\prime }$ corresponding to the channel is divided into a group to form 64 groups of feature blocks $B_{q}^{\prime \prime }$ (7 × 7 × 1, n); q depicts the $q^{\mathrm{th}}$ layer. The spatial weight matrix $W_{q}^{s}$ is determined by the similarity between sub-feature blocks in $B_{q}^{\prime \prime }$.
(15)$$\begin{array}{c} W_{q}^{s}=softmax\left(SM^{\prime }\left(B_{q}^{\prime \prime }\right)\right),\left(q=1,2,\ldots ,64\right) \end{array}$$
$B_{q}^{\prime \prime }$ is regarded as a matrix with 1 × n, multiplied by $W_{q}^{s}$ to form a feature map with w × h × 1. Finally, all channels are merged to form feature map $F_{1.2}$.
(16)$$\begin{array}{c} F_{1.2}=concat\left(B_{q}^{\prime \prime }\times W_{q}^{s}\right) \end{array}$$

## 3. Experimental Results and Analysis

### 3.1. Implementation Details

#### 3.1.1. Dataset

##### Deep Learning Dataset

Five hundred clean infrared images are randomly selected from the infrared image dataset LTIR v1.0 [27]. These images are cropped into 49 × 49 image patches, and the data augmentation methods (symmetric flip, rotation and scale) are used to expand the number of image patches. Then, 200,000 image patches are generated. The datasets are divided into training, validation and test datasets, which include 196,000, 2000 and 2000 images, respectively.

In a real scene, the intensity of stripe noise is not constant. Hence, by adding non-uniformity noise with mean 0 and standard deviation from 0 to 0.15 to the training dataset, the model could learn to handle stripes of different intensities.

##### Experimental Dataset

For network analysis and the simulated noise dataset, non-uniformity noise with mean 0 and standard deviation of 0.01, 0.02, 0.03, 0.05 and 0.10, respectively, is manually added to 20 clean infrared images from DLS-NUC [16].

The real noise dataset is 20 images from the public infrared dataset on the internet [28].

#### 3.1.2. Loss Function

As we all know, L1 and L2 are widely used loss functions in the field of image restoration. However, compared with L2, L1 has better correlation in the qualitative and quantitative evaluation of image quality [29,30]. Consequently, L1 is used as the loss function; its expression is the mean square error between estimated stripe noise Inoise^ by model training and real stripe noise $I_{noise}$ in the image, as depicted in:(17)Loss=‖Inoise−Inoise^‖1
where ${\left\|\cdot \right\|}_{1}$ is the 1-norm.

#### 3.1.3. Training

In the training stage, the proposed model is trained 50 epochs using the adaptive moment estimation (ADAM) optimization method [31] with mini batch 128, to optimize the loss function. The initial learning rate is set to 0.001 and then decreased by the factor of 10 every 25 epochs. The ‘he_normal’ [32] is used to initialize the network parameters.

All experiments are carried out in the Tensorflow 2.5 environment and run on two NVIDIA 3060Ti GPUs.

#### 3.1.4. Comparing Approaches

The proposed method is compared with four single-framed de-stripe methods, including 1-d guided filtering (1DGF) [33], SNRCNN [15], DLS-NUC [16] and ICSRN [17]. The source codes of these methods are publicly available.

### 3.2. Network Analysis

#### 3.2.1. Multi-Scale Representation

In order to verify the effectiveness of MFE, we compare MFE with conv1-3(1 × 1 + 3 × 3), conv1-3-5(1 × 1 + 3 × 3 + 5 × 5) and conv1-3-5-7(1 × 1 + 3 × 3 + 5 × 5 + 7 × 7) convolution filters on the same dataset.

Figure 5 shows the performance of different convolution filters on the test set. Our proposed structure achieves higher peak signal-to-noise ratio (PSNR) and faster convergence, which shows that MFE is adept in using image information.

#### 3.2.2. Attention Mechanism

To demonstrate the effectiveness of CSAS, we train the network with CSAS, channel attention mechanism based on similarity (CAS), spatial attention mechanism based on similarity (SAS), and without attention mechanism. Performance curves are exhibited in Figure 6.

Evidently, CAS and SAS have higher PSNR than without the attention mechanism, which reflects the effectiveness of the attention mechanism. The channel spatial attention mechanism based on similarity reaches a higher performance, compared with SCA and SSA. Such a result demonstrates that CSAS effectively extracts image features in both channel and space, which is more conducive to separating stripe noise and scene details.

### 3.3. Experiments with Simulated Noise Infrared Images

Noise intensity determines algorithm performance. The higher the stripe noise intensity, the more difficult it is for the algorithm to accurately remove stripe. Through the experiment, it is found that images with noise intensity above 0.05 have dense stripes, which is enough to verify the algorithm performance. Thereby, stripe noise with different intensities (0.01, 0.02, 0.03, 0.05 and 0.10) is manually added to the clean infrared image for experiment.

#### 3.3.1. Qualitative Evaluation

The qualitative evaluation is the visual perception. The visual effect of removing stripe noise with different intensity is illustrated in Figure 7. With the increase in stripe noise intensity, the performance of other methods decreases significantly. The residual stripe will appear in images. However, our method is hardly affected by the noise intensity and completely clears most of the stripe noise.

Figure 8 illustrates the denoising effect of each algorithm upon images with non-uniform noise intensity of 0.03. The ability of DLS-NUC and ICSRN to erase stripe noise is relatively weak. We can clearly observe some residual stripe noise. 1DGF and SNRCNN show a better stripe removal effect, but there is still some residual stripe. Significantly, our method achieves a remarkable de-striping result. The stripe is smoothed away, and the detail is retained to the maximum extent. That is because the proposed model learns the stripe property with different intensities in the training stage; it can adaptively remove the stripe noise in the image.

#### 3.3.2. Quantitative Evaluation

In the experiment of simulated noise infrared images, two common full reference indicators for image evaluation (PSNR [34] and SSIM) are applied to evaluate the de-striping performance.

PSNR: reflects the error between the two images. The larger the value, the smaller the distortion.

SSIM: reflects the degree to which the original image details are preserved. The larger the value, the more accurate the preserved details.

The mean values of the PSNR and SSIM indices for each method are listed in Table 1. The best results for each noise intensity are highlighted in bold. The mean PSNR and SSIM values of all methods significantly decrease with the increase in noise intensity. In contrast to the comparative methods, our method achieves stable de-striping performance against the pattern noise strength, where the mean PSNR and mean SSIM are over 40.08 dB and 0.98, severally. This shows that our method is suitable for images with varying degrees of stripe noise.

For 100 simulated infrared images with different noise intensities, Figure 9 and Figure 10 represent the PSNR and SSIM of different stripe removal methods. It is noticed that our method achieves relatively high PSNR and SSIM, and the corrected image is closer to the original image.

### 3.4. Experiments with Real Noise Infrared Images

#### 3.4.1. Qualitative Evaluation

The corrected results for real noise infrared images with rich details are illustrated in Figure 11. 1DGF has a good stripe removal effect, but a certain amount of detailed information is lost. SNRCNN and ICSRN can protect the details and edge information of the image, but it still has obvious stripe noise. DLS-NUC fails to simultaneously balance the stripe noise and details, the branches are blurred, and the stripe noise still exists. In comparison, our method retains the details of the image while removing the stripe noise. There is no stripe noise in Figure 11f, and the texture information of the branches and leaves is well saved.

The corrected results for the real noise infrared images with vertical edge are exhibited in Figure 12. For 1DGF, although the stripe noise is eliminated, the entire image becomes blurred. SNRCNN incorrectly extends and blurs the edge information of the building. The correction result of DLS-NUC produces ghosting artifacts in the target position with vertical edges. There is still a small amount of stripe noise in the correction result of ICSRN. The method we proposed removes the stripe noise without producing any ghosting artifacts, avoids judgment of strong stripe noise as edge, and balances well between NUC and vertical edge information preservation.

The corrected results for the real noise infrared images with more intense stripes are exhibited in Figure 13. 1DGF achieves a better de-striping effect, but has some detail loss. There is still some obvious stripe noise in SNRCNN and ICSRN. DLS-NUC blurs the image details while blurring stripe. The method we proposed erases the stripe noise and hardly loses the details.

The NUC results of some other different image scenes are shown in Figure 14. It can be seen that the correction results of the five algorithms are evidently different. The proposed method achieves a good visual effect in all image scenes.

To further prove the effectiveness of the proposed method, taking the original infrared image of Figure 11a as an example, we calculated the column mean of the original image and the corrected images. The result is shown in Figure 15. The original image has large fluctuations in the column average curve. ICSRN still has large fluctuations. SNRCNN and DLS-NUC diminish the fluctuations, but there are still small fluctuations, indicating uncorrected residual non-uniformity. 1DGF eliminates these small fluctuations, but is too smooth (such as at the corner of a curve), which can cause loss of image detail. The proposed method not only smooths the stripe noise, but also preserves the detailed information of the image (such as the corner of the curve).

#### 3.4.2. Quantitative Evaluation

In order to further verify the performance of the proposed method, a non-reference indicator (roughness) [35,36] is used for quantitative evaluation in the experiment of real noise infrared images.

Table 2 shows roughness of images corrected by different methods. From the quantitative evaluated results, the proposed method outperforms the other four NUC methods.

Figure 16 depicts the quantitative evaluation results of 20 real noise infrared images corrected by different methods. Compared with the other methods, the proposed method has smaller roughness and more effectively suppresses the non-uniformity of the image.

## 4. Conclusions

In this paper, a NUC method for a single infrared image based on a multi-scale attention mechanism is proposed, which utilizes residual strategy to learn the stripe features. The MFE model is utilized to extract various coarse and fine features. Through the similarity of feature map blocks, the CSAS model can adaptively filter out useful information, separate the scene details and stripe features more thoroughly and further improve the representational ability of the network. Compared with four state-of-the-art methods, our proposed approach shows a sharper visual effect without perceptible ghosting artifacts. The simulated noise images validate that our approach is robust and can remove stripe noise with diverse intensities. The real noise images test and verify that our approach has better detail retention, less noise residue, and effectively separates stripe noise and edge information.

## Figures and Tables

**Figure 1 sensors-22-08734-f001:**
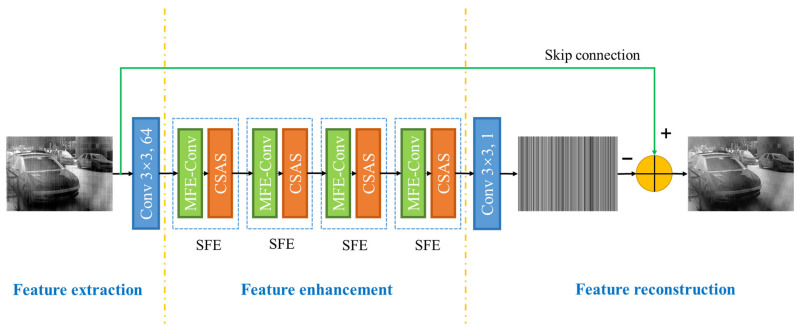
The network architecture of the proposed method.

**Figure 2 sensors-22-08734-f002:**
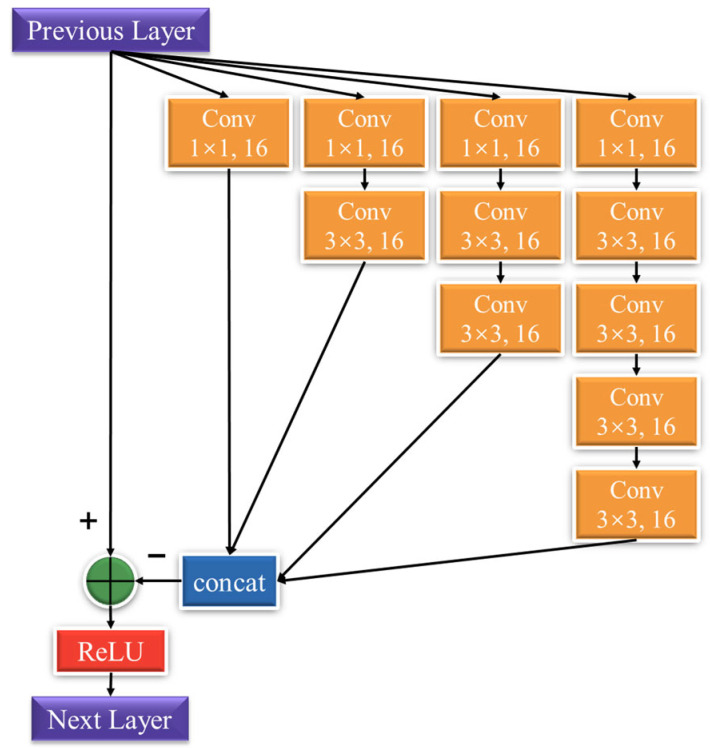
The structure of MFE.

**Figure 3 sensors-22-08734-f003:**
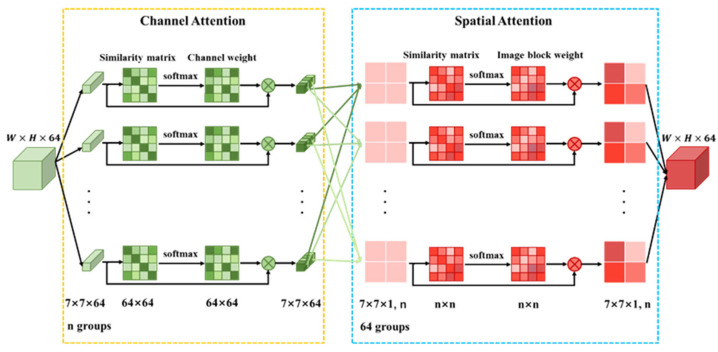
The detail architecture of CSAS.

**Figure 4 sensors-22-08734-f004:**
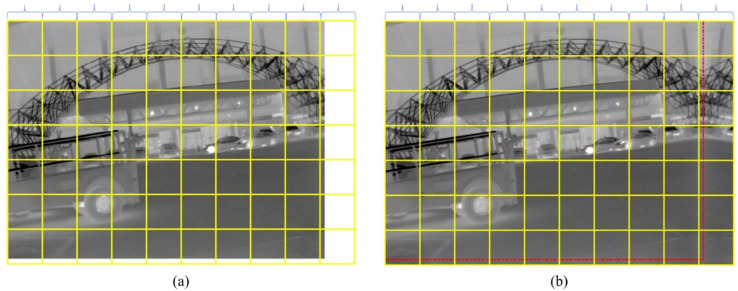
Image block division. (**a**) The raw image; (**b**) The filled image.

**Figure 5 sensors-22-08734-f005:**
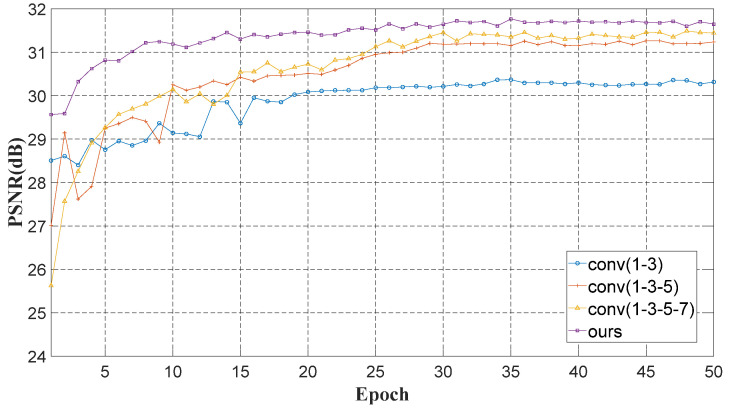
PSNR curves of various convolution filters.

**Figure 6 sensors-22-08734-f006:**
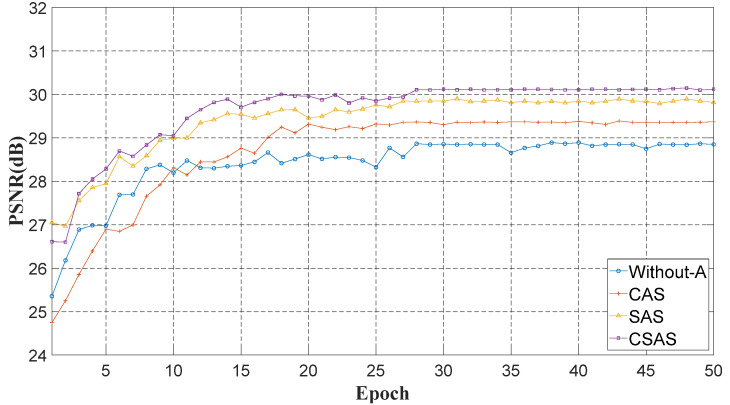
PSNR curves of various attention mechanisms.

**Figure 7 sensors-22-08734-f007:**
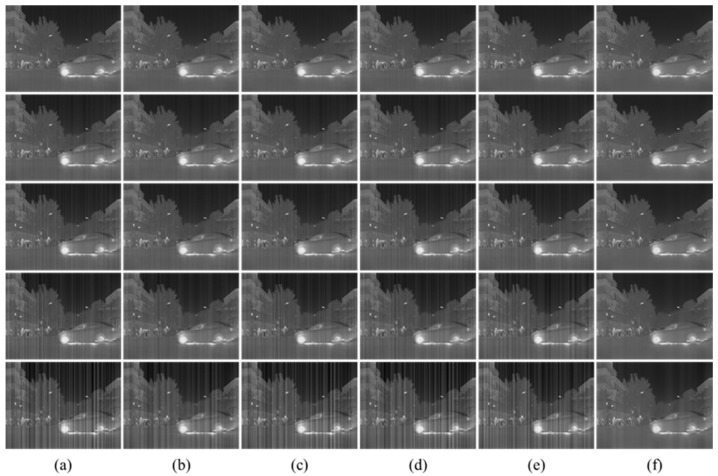
The NUC results of different methods for different intensity noise. (**a**) The noise infrared images with different intensity (the noise intensity is 0.01, 0.02, 0.03, 0.05, 0.10 from top to bottom); (**b**) 1DGF [33]; (**c**) SNRCNN [15]; (**d**) DLS-NUC [16]; (**e**) ICSRN [17]; (**f**) Our method.

**Figure 8 sensors-22-08734-f008:**
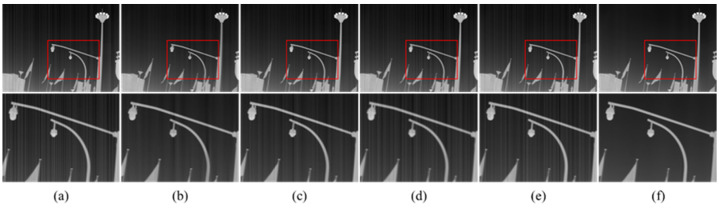
The NUC results of different methods. Top: raw infrared image and the NUC results; Bottom: Zoom-in views on the highlighted area. (**a**) The noise infrared images with intensity of 0.03; (**b**) 1DGF [33]; (**c**) SNRCNN [15]; (**d**) DLS-NUC [16]; (**e**) ICSRN [17]; (**f**) Our method.

**Figure 9 sensors-22-08734-f009:**
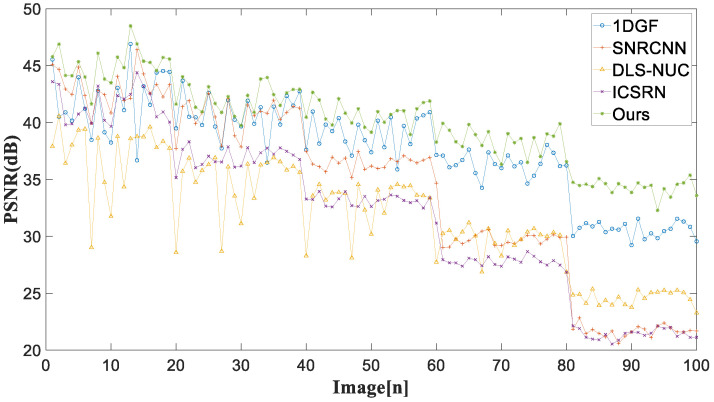
PSNR (dB) of different methods for 100 simulated noise infrared images.

**Figure 10 sensors-22-08734-f010:**
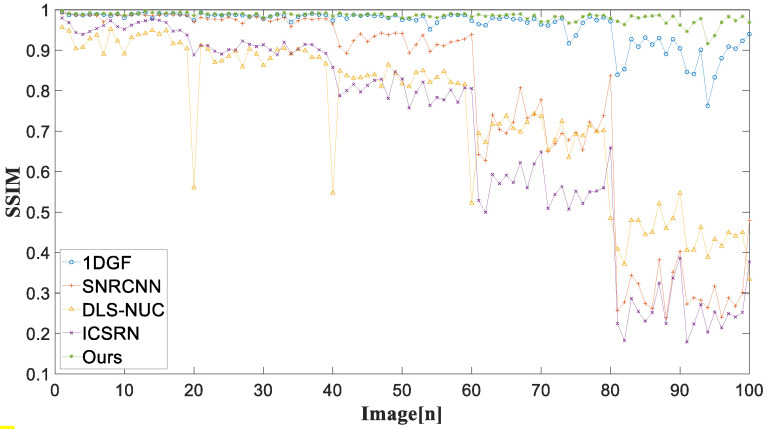
SSIM of different methods for 100 simulated noise infrared images.

**Figure 11 sensors-22-08734-f011:**
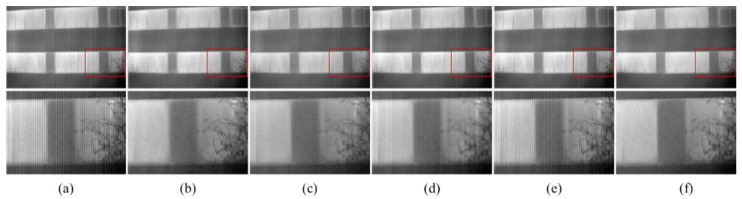
The NUC results of different methods. Top: raw infrared image and the NUC results; Bottom: Zoom-in views on the highlighted area. (**a**) The raw images; (**b**) 1DGF [33]; (**c**) SNRCNN [15]; (**d**) DLS-NUC [16]; (**e**) ICSRN [17]; (**f**) Our method.

**Figure 12 sensors-22-08734-f012:**
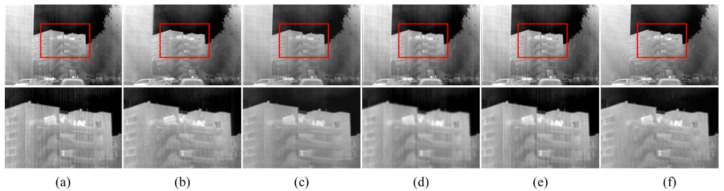
The NUC results of different methods. Top: raw infrared image and the NUC results; Bottom: Zoom-in views on the highlighted area. (**a**) The raw images; (**b**) 1DGF [33]; (**c**) SNRCNN [15]; (**d**) DLS-NUC [16]; (**e**) ICSRN [17]; (**f**) Our method.

**Figure 13 sensors-22-08734-f013:**
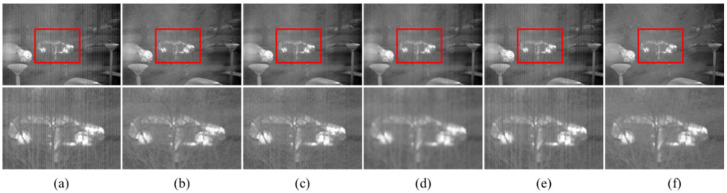
The NUC results of different methods. Top: raw infrared image and the NUC results; Bottom: Zoom-in views on the highlighted area. (**a**) The raw images; (**b**) 1DGF [33]; (**c**) SNRCNN [15]; (**d**) DLS-NUC [16]; (**e**) ICSRN [17]; (**f**) Our method.

**Figure 14 sensors-22-08734-f014:**
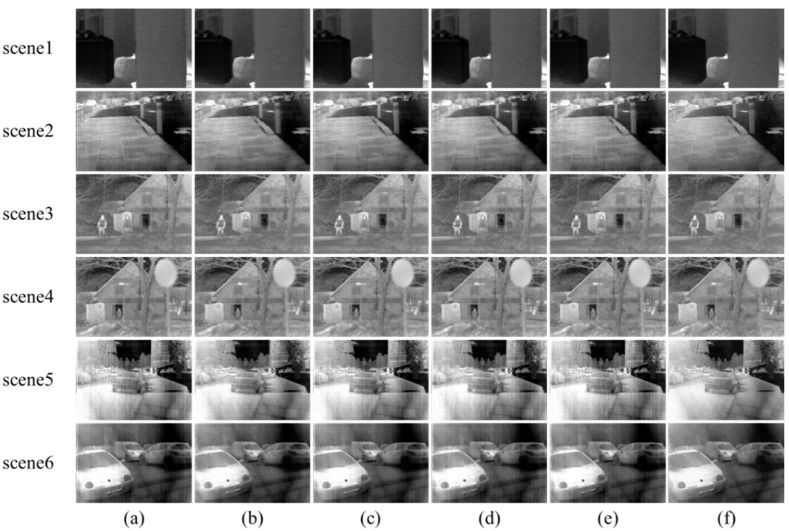
The NUC results of different methods. (**a**) The raw images; (**b**) 1DGF [33]; (**c**) SNRCNN [15]; (**d**) DLS-NUC [16]; (**e**) ICSRN [17]; (**f**) Our method.

**Figure 15 sensors-22-08734-f015:**
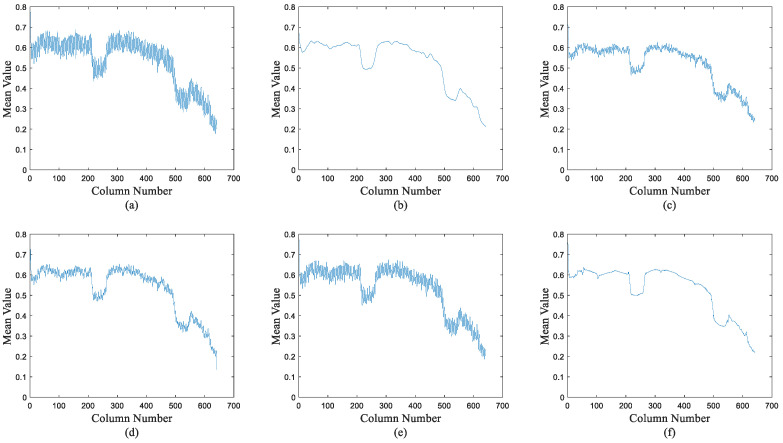
Column mean transformation curves of original and corrected images. (**a**) The raw images; (**b**) 1DGF [33]; (**c**) SNRCNN [15]; (**d**) DLS-NUC [16]; (**e**) ICSRN [17]; (**f**) Our method.

**Figure 16 sensors-22-08734-f016:**
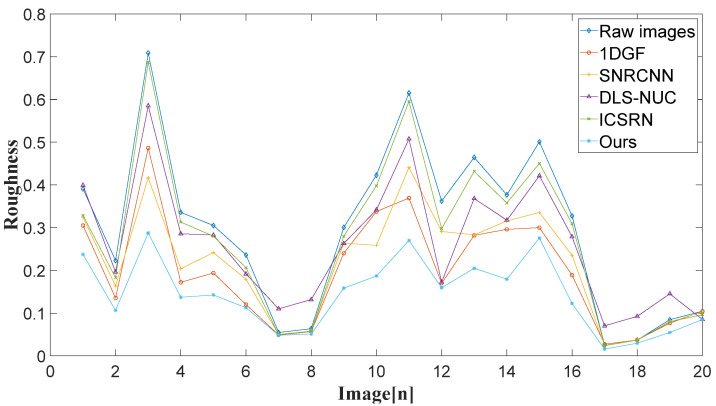
Roughness of different methods for 20 real noise infrared images.

**Table 1 sensors-22-08734-t001:** Mean PSNR (dB)/SSIM results of different methods on 100 simulated noise infrared images.

Noise Intensity	Methods				
	1DGF [33]	SNRCNN [15]	DLS-NUC [16]	ICSRN [17]	OURS
**0.01**	41.8133/0.9876	42.8198/0.9858	36.8545/0.9094	41.1695/0.9562	**44.9701/0.9916**
**0.02**	40.5641/0.9851	40.5567/0.9755	34.8792/0.8707	36.8761/0.9030	**42.0900/0.9889**
**0.03**	39.0406/0.9808	36.3485/0.9232	32.9968/0.8173	33.0467/0.8006	**40.6686/0.9866**
**0.05**	36.3567/0.9688	29.7059/0.7116	29.7599/0.6909	27.7736/0.5664	**38.3707/0.9822**
**0.10**	30.5887/0.8885	21.6873/0.3058	26.6098/0.4417	21.4158/0.2584	**34.3057/0.9697**

**Table 2 sensors-22-08734-t002:** Roughness index (ρ) on real noise infrared images.

Image	Raw Image	Methods				
		1DGF [33]	SNRCNN [15]	DLS-NUC [16]	ICSRN [17]	OURS
Figure 5	0.3360	0.1722	0.204	0.2858	0.3133	**0.1372**
Figure 4	0.3050	0.1939	0.2418	0.2828	0.2800	**0.1426**
Figure 12	0.3621	0.1701	0.2911	0.1728	0.2978	**0.1595**

## Data Availability

Not applicable.

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
