# Peer review of "Single Infrared Image Stripe Removal via Residual Attention Network"

_sensors, 2022, doi:10.3390/s22228734_

Round 1

Reviewer 1 Report

1. Despite the fact that the title is good, it is incomplete. I would appreciate it if you could provide a word that describes the application. For example, what does it do?

2. What is the best way to validate these statistical results given in the abstract "The average peak signal-to-noise ratio and average structural similarity of the infrared image corrected by applying our method exceed 40.08 dB and 0.98, respectively." Please provide verification of these values in the paper.

3. It would be a good idea to put the solution based on the problem in the introduction.

4. As mentioned in the introduction, the contribution should be based on the problems mentioned in the introduction.

5. In the absence of literature concerning the Attention Network, which network is used in order to implement it? 

6. Figure 1. It consists of three parts, namely, feature extraction, feature enhancement, and feature reconstruction. Why there is also a forward path in the diagram, but it is not explained in any way.

7. Figure 4. image is used from some source, please use the image from the dataset or image used in the results.

8. According to figure 2, in the MFE, only 1x1+3x3 are used, but in the results, 7(1x1+3x3+5x5+7x7) are used, can you please confirm this?

9. In the Table 1 the noise intensity is mentioned, can you tell me on what basis you chose these intensities, and why did your results improve when the intensity is 0.03?

10. As shown in Figures 5 and 6 is PNSR vs epoch, where the graph shows an increase when it comes to number of images, while in Figure 9 the graph indicates a decrease when it comes to number of images, so please double check the graph.

11. As shown in Figure 15, there is no difference between b and f. 

12. In the conclusion, it should be concluded that the statical results have been obtained

Author Response

I would like to thank you for giving me an opportunity to revise my manuscript. I appreciate you very much for the positive and constructive comments on my manuscript. These comments are all valuable and helpful for revising and improving the manuscript. I have carefully studied these comments and have made corrections.Please see the attachment.

Reviewer 2 Report

The authors worked on the correction of infrared images in terms of reducing non- uniformity by proposing a residual attention network. I have the following major comments:

1.       It is very difficult to follow the paper. Therefore, kindly use different symbols/notations for the matrixes, scalars and vectors in the whole paper.

2.       Although the accuracy is enhancing in the proposed methodology, a comparison between the proposed and existing techniques in terms of computational complexity would be better. The authors can use lamda big O notation to compare the complexities.

3.       In the comparison tables and figures, instead of only writing the technique name, it would be further better if the authors could put the relevant reference as well.

4.       It is very difficult to read the Figure 15. Therefore, please enhance the quality of that figure.

5.       The reviewer finds that the proposed methodology has been widely discussed in the literature:

Juntao Guan, Rui Lai, Ai Xiong, Zesheng Liu, Lin Gu, Fixed Pattern Noise Reduction for Infrared Images Based on Cascade Residual Attention CNN, Neurocomputing (2019), doi: https://doi.org/10.1016/j.neucom.2019.10.054

Non-Uniformity Correction of Infrared Images Based on Improved CNN With Long-Short Connections Volume 13, Number 3, June 2021

A. Kujur, Z. Raza, A. A. Khan and C. Wechtaisong, "Data Complexity Based Evaluation of the Model Dependence of Brain MRI Images for Classification of Brain Tumor and Alzheimer’s Disease," in IEEE Access, vol. 10, pp. 112117-112133, 2022, doi: 10.1109/ACCESS.2022.3216393.

       6.  What is the difference between the proposed methodology and the above-mentioned references?

       7. The contributions of the paper are not sufficient for the acceptance in a prestigious journal, such as sensors. Therefore, it is recommended to explain the contributions in further details.

Author Response

I would like to thank you for giving me an opportunity to revise my manuscript. I appreciate you very much for dedicated review and invaluable comments on my manuscript. These comments are all valuable and helpful for revising and improving the manuscript. I have carefully studied these comments and have made corrections. Please see the attachment.

Round 2

Reviewer 2 Report

The authors have significantly revised the manuscript compared to the previous version by following the reviewers comments. Therefore, the paper can be accepted now.